Blood circulation in the ascidian tunicate Corella inflata (Corellidae)

Konrad Michael W. michael@scienceisart.com
Sausalito , CA , United States
Wang Lingling
Electronic publication date: 2016 Dec 14
Publication date: 2016
Volume: 4
Electronic Location ID: e2771
Received 2016 Jun 11; Accepted 2016 Nov 8
Copyright: ©2016 Konrad
Copyright year: 2016
Copyright holder: Konrad
License: This is an open access article distributed under the terms of the Creative Commons Attribution License, which permits unrestricted use, distribution, reproduction and adaptation in any medium and for any purpose provided that it is properly attributed. For attribution, the original author(s), title, publication source (PeerJ) and either DOI or URL of the article must be cited.
License URL: https://creativecommons.org/licenses/by/4.0/

Keywords: Heart, Branchial basket, Blood velocity, Closed circulation, Blood circulation, Pharynx, Closed

Funding: The author received no funding for this work.

==============================
The body of the ascidian tunicate Corella inflata is relatively transparent. Thus, the circulatory system can be visualized by injecting high molecular weight fluorescein labeled dextran into the heart or the large vessels at the ends of the heart without surgery to remove the body wall. In addition, after staining with neutral red, the movement of blood cells can be easily followed to further characterize the circulatory system. The heart is two gently curved concentric tubes extending across the width of the animal. The inner myocardial tube has a partial constriction approximately in the middle. As in other tunicates, the heart is peristaltic and periodically reverses direction. During the branchial phase blood leaves the anterior end of the heart by two asymmetric vessels that connect to the two sides of the branchial basket. Blood then flows in both transverse directions through a complex system of ducts in the basket into large ventral and dorsal vessels which carry blood back to the visceral organs in the posterior of the animal. During the visceral phase blood leaves the posterior end of the heart in two vessels that repeatedly bifurcate and fan into the stomach and gonads. Blood velocity, determined by following individual cells in video frames, is high and pulsatory near the heart. A double peak in velocity at the maximum may be due to the constriction in the middle of the heart tube. Blood velocity progressively decreases with distance from the heart. In peripheral regions with vessels of small diameter blood cells frequently collide with vessel walls and cell motion is erratic. The estimated volume of blood flow during each directional phase is greater than the total volume of the animal. Circulating blood cells are confined to vessels or ducts in the visible parts of the animal and retention of high molecular weight dextran in the vessels is comparable to that seen in vertebrates. These are characteristics of a closed circulatory system.

Introduction

Vertebrates are one of three subphyla in the phylum Chordata. Genomic DNA sequence analysis strongly suggests that the tunicate subphyla, not the apparently more similar cephalochordates are the sister group to vertebrates (Delsuc et al., 2006). Thus, the study of tunicates could play an important role in understanding vertebrate evolution from a common ancestor.

Ciona intestinalis, a member of the class Ascidiae, has been used in many of the recent studies of embryological development and gene expression, (Stolfi & Christiaen, 2012) and it is the subject of many hundreds of publications. In particular, the development of the heart (Christiaen et al., 2009; Davidson, 2007; Davidson et al., 2006; Stolfi et al., 2010), and by association the circulatory system, has been the subject of recent reports. The three orders of Ascidiae are differentiated by the morphology of the branchial basket that supports the filter feeding net. Both C. intestinalis and C. inflata are in the order Phlebobranchiata, with a basket vasculature intermediate in complexity in the three orders. However, C. intestinalis is in the family Cionidae, with visceral organs posterior to the branchial basket. Berrill (1955) considers this family primitive because it is “the only family in which viscera continue to occupy the position they held in the beginning (of morphogenesis), …in all other families the viscera either descends into the stalk or is dislocated along one side of the branchial region.” C. inflata, the tunicate described in this study, (Lambert, Lambert & Abbott, 1981) is in the family Rhodosomatidae, with a digestive track and heart on the right side of the body (van Name, 1945).

In tunicates motion of the peristaltic heart is often visible, and it is apparent that the direction of pumping reverses periodically, typically every one to three minutes. At present there is no commonly accepted physiological advantage for the reversal, but it may be at the root of the suspicion that the circulatory system in tunicates is fundamentally different from the vertebrates in which blood always flows in one direction. For example, the heart might cycle in direction to pump blood from one large internal sinus to the other. However, Kriebel (1968) estimated the total volume of blood pumped during one directional phase as the product of the volume of the heart and the number of heart beats, and found this volume to be equal to the weight of the animal, far too large to be stored in any internal sinus. This same publication mentioned several citations presenting evidence that in one directional phase the blood completes several complete circulations, and describe the system as closed. There is no experimental evidence for lack of a complete circulation loop from one end of the heart to the other.

However, circulation in tunicates is often described as “open” (Davidson, 2007; Monniot, Monniot & Laboute, 1991; Passamaneck & Di Gregorio, 2005; Satoh, 1994). An “open circulation” usually refers to a specific configuration often seen in crustaceans and insects. Blood is pumped through a vascular network that delivers it to various parts of the body. The blood then percolates around cells of the body tissues to finally collect in a large, pericardial sinus. The heart siphons blood from this sinus to complete the circulation. In an open circulation the term “hemolymph” is preferred over “blood” because the fluid is in direct contact with somatic cells during a large portion of its circulation, and thus is more analogous to lymph than the blood in the closed vertebrate circulation. However, there has been no report of a pericardial blood sinus in any ascidian.

The arguably most complete description of the anatomy of an ascidian tunicate is the 123 page monograph Ciona, (Millar, 1953), where the circulatory system is described in detail. The ventral end of the heart connects directly to the vessel that runs under the ventral edge of the branchial basket. Blood then flows through many parallel transverse vessels across the branchial basket to empty into the large dorsal vessel and flow back to the visceral, posterior region of the animal. Since sea water flows across both sides of the branchial basket, blood must always be enclosed, or it would be lost. The ventral, dorsal end of the heart is connected to a bifurcating vessel tree supplying blood to the stomach, ovaries, testes, etc. There is no mention of a pericardial sinus in Millar’s work. In some publications circulation in tunicates is said to be open because the blood vessels lack endothelial cells. Citations are not given, but the source is again probably Millar (1953). In this publication reference to endothelial cells and blood vessels consists of one sentence in which the author states that endothelial cells were only found associated with vessels near the heart. There is no mention of criteria used to define endothelial cells, or which vessels in the animal were studied. Thus documentation of the histology of tunicate vessels is rather thin, and certainly deserves more attention. However, an anatomically closed circulatory system which is described as open only because the vessels are permeable or leaky has not been reported in the literature.

The present report is the first description of the circulatory system of the ascidian tunicate Corella inflata (Lambert, Lambert & Abbott, 1981). This animal is more transparent than C. intestinalis, and the circulatory system is clearly visible without the need of surgery to remove the tunic and body wall. Staining with neutral red dye facilitated observation of moving blood cells which defines the circulatory system. In contrast to C. intestinalis blood leaves the ventral end of the heart in two vessels which supply the two sides of the branchial basket. Individual cells could not be followed in the peristaltic heart but blood flow and motion of the myocardial heart tube suggests an attachment at a constriction in the middle of the heart.

Measurement of blood cell positions in a series of consecutive video frames allowed construction of graphs of cell velocity versus time near the heart. Blood flow is pulsating, falling almost to zero between a double maximum, which may be a consequence of the constriction in the middle of the heart tube.

The body of C. inflata is rigid, and does not contract significantly when probed with needles. This insensitivity facilitated injection of high molecular weight dextran into the circulatory system at both ends of the heart and during both directional phases of heart action. Distribution of dye clearly reveals two bifurcating trees of vessels rooted at both ends of the heart. Retention times of the dye in vessels were similar to that seen in vertebrates, and thus the vessels are no more permeable than vertebrate vessels, at least at the blood pressures present in the tunicate.

Methods

Animal collection and care

Corella inflata Huntsman 1912 (Lambert, Lambert & Abbott, 1981) were collected in a marina in Sausalito, CA, USA (San Francisco Bay) from the side of a floating dock at a depth of 0.1–0.3 m. The present study is the result of observations on 94 animals over a period of 15 months.

Tunicates were kept in a seawater aquarium at temperatures of 16–20C with aeration, and used within 72 h of collection. During injections and observations animals were submerged in seawater and pinned to a 5 mm thick gel of silicon (Slygard 184; Global-Industrial Corp) in a 7 × 7 cm acrylic box. The top of the animal was typically only 1-3 mm below the surface of the seawater. No surgery was done to remove the tunic or any other part of the animal.

Staining with neutral red

Tunicates were stained by immersion in 50 mL of a 0.1% solution of neutral red dye (Cynmar Corp.) in seawater for 10 min. They were then washed twice in 50 mL of seawater.

Injection of fluorescent dextran

Fluorescent dextran (FITC labeled, 150,000 Daltons; Polysciences, Inc.) at a concentration of 5 mg/mL in seawater was injected using a 33-gauge × 1/2 inch hypodermic needle (TSK STERiJECT PRE-33013) connected by 8 cm of polyethylene tubing (PE/8, 1.2 mm ID; Scientific Commodities Inc.) to a 1 mL plastic tuberculin syringe (102-ST1C; McKesson) with a total plunger travel of 57 mm. The syringe plunger was directly connected to a screw attached to a microprocessor controlled stepping motor with 200 steps per revolution. Rotation of the screw thus rotated the plunger to overcome static friction that could affect fluid delivery. One revolution advanced the screw 0.85 mm corresponding to 15 uL. Injections were controlled by joystick, with stepper motion recorded to a computer file. Typically, 5–20 uL were injected over a 3 s to 15 s period.

Imaging of fluorescent dextran

Excitation was accomplished using a single LED (Luxeon V Star Blue, 470 nm, 48 lm) placed about 40 mm from the animal. Fluorescence was imaged using a long wavelength pass filter Schott GG495, (UQG Optics Limited) to block reflection of the excitation LED.

Image capture and processing

A Canon Rebel T3i camera and Canon 100 mm macro lens mounted on a copy stand were used to obtain images of the entire animal. Higher magnifications, required to follow moving blood cells, was obtained using a Meiji stereoscopic microscope and the Canon camera. Images were processed using Adobe Photoshop software with any image enhancement applied uniformly to the entire frame. The velocity of blood cells was determined by measuring position of a selected cell in sequential frames from a video. Video files were converted to individual images using QuickTime Player 7 (Apple) and the positions of cells were determined using Photoshop (Adobe). The default configuration of Photoshop gives x,y positions to 0.1 pixel, which is useful when one wants to specify the center of an object of 3–16 pixels. There are many other combinations of applications that could obtain similar data. In this report velocity is defined as distance/time, where time is 1/30 s, the time between each video frame, and data in graphs representing a moving average over 5 time points.

Results

Transparency of C. inflata facilitates visualization of internal structure if the structure is not itself transparent. Thus, optimal observation is achieved by using a dye to create contrast. Neutral red is most commonly used in tissue culture as a stain specific for living cells, where it is transported into lysosomes to give the culture a red-pink color (Borenfreund & Puerner, 1984), while also staining nuclear chromatin (Espelosin & Stockert, 1982). Because it is non-toxic and bound complexes are stable, it has been used to stain entire marine invertebrates for up to seven days (Drolet & Barbeau, 2006). In the present study, neutral red was used as a relatively non-specific stain for blood cells which were identified by motion. Moving blood cells define the vessels, channels or ducts that distribute blood, and the velocity of the cells reveals the dynamics of the circulatory system. Alternatively, injection of fluorescent dextran into the circulatory system while the heart is pumping defines the distribution of blood throughout the vascular system.

In this report, the terms “vessel” and “vascular” are used only to specify pathways, channels or ducts through which blood flows. There is no intent to suggest that a “vessel” has the histology of a blood vessel in a vertebrate, as the methods and equipment used in this study do not resolve structure at this scale.

Structure of the tunicate and location of major blood vessels

The right and left sides of a typical C. inflata stained with neutral red are seen in Figs. 1A and 1B respectively, with labeled diagrams below each photograph. Tunicates have an inner cylindrical branchial basket attached along the ventral edge, the endostyle, to the outer cylindrical body. Water is pumped into the oral siphon and out of the atrial siphon by cilia in the branchial basket. Food is trapped in a mucus net produced by the endostyle, moved by ciliary action across the interior of the branchial basket and then drawn down into the stomach. C. inflata has an enlarged exit siphon, which in this image contains fecal matter, but is also used to contain and protect the tunicate “tadpoles.” The endostyle is the dark band along the ventral side of the animal, the orange band in the diagram. The stomach, ovaries, and testis make up the dark mass at the bottom of the tunicates. The heart is the vertical band along the upper edge of the visceral mass.

Figure 1 Tunicate stained with neutral red.

(A) Right side of animal; ventral at right, dorsal left, anterior top, posterior bottom. Scale bar equals 10 mm. (B) Left side; animal rotated 180°about vertical axis. Abbreviations: atrial, atrial siphon; oral, oral siphon; dbv, dorsal branchial vessel; vbv, ventral branchial vessel; orange strip, endostyle; arrows indicate blood flow during the branchial phase.

In C. inflata two large blood vessels exit the anterior (ventral) end of the heart and pass to the middle of each side of the branchial basket where they repeatedly branch and merge into a rectangular net of vessels. One vessel connects directly to the middle of the right half of the basket, Fig. 1A, while the other vessel passes over the endostyle and connects to the left side of the branchial basket, Fig. 1B. The approximate gross volume (all interior space including sea water) of the tunicate is estimated to be 2,700 mm3 from the photograph and assuming the animal is a solid trapezoid with constant thickness.

Global pattern of blood flow

Moving blood cells define the functional vessels (Fig. 2 and Video S1). The peristaltic heart of C. inflata, like other tunicates, reverses the direction of pumping every few minutes. During the branchial phase of heart action blood is drawn from the viscera into the posterior end of the heart and then pumped from the anterior end into the branchial basket via two vessels. One vessel leaves the heart and is seen to join the branchial basket on the right side of the animal but the vessel that carries blood to the left side of the branchial basket cannot be seen in the perspective of Fig. 2. Blood that enters the middle of the right side of the branchial basket flows in a rectilinear network to vessels at both the dorsal and ventral edges of the basket, and then flows in a posterior direction toward the viscera to complete the circulation. A small fraction of blood leaving the heart supplies the mantle and tunic, but this vessel system is not visible in the focal plane of Fig. 2 and Video S1.

Figure 2 Flow of stained blood cells in the right side of the tunicate.

The ventral side of the tunicate is down and the anterior end is to the right. Blood leaves the heart at its ventral end and immediately splits into two vessels. Only the one supplying the right side of the branchial basket is shown here. After flowing through the branchial basket, blood is collected by ventral and dorsal vessels and flows in the posterior direction into the viscera (green area marked by “V”). Blood is collected from the viscera in a network of small vessels that progressively merge to form two large vessels that enter the heart at its dorsal end. When the heart pumps in the posterior direction all flows are reversed. In the simplified diagram only the outer heart tube is drawn, and thus there is no constriction. Scale bar at lower left is 10 mm.

Video S1 demonstrates the ability of blood cell movement to identify the circulation system but it only provides information about a limited region of the animal. In order to build the more complete model of circulation described in the text it was necessary to focus at different planes, examine various fields of view, rotate the animal to see regions that are obscured by stained structures, and examine many animals. A dramatic feature of Video S1 is the near synchronous movement of pulsating blood cells throughout the animal as the heart beats. This implies both that the output of the heart is pulsatory and that the phase is approximately constant throughout the animal.

Injection of fluorescent dye into the circulatory system

Injection of the non-binding tracer, fluorescent dextran, into the circulation allows direct visualization of the heart and the larger vessels. However, the distribution of viscera is laterally asymmetric, and hides much of the circulation in the left side of the tunicate. Thus only the right side is shown in this study. Dextran was injected into the posterior end of the heart pumping in the branchial phase. The dye then flowed into a vessel leaving the anterior end of the heart and into the right side of the branchial basket where it filled a rectilinear vascular network (Fig. 3A). Dextran flowing to the left half of the branchial basket cannot be seen in this view.

Figure 3 Florescent dextran injected into the circulatory system.

(A) Dextran injected (25 uL over 1.9 min) at the posterior end of heart while the heart pumped in the branchial direction. This image was obtained 3.5 min after end of injection. Dextran enters the branchial basket through a branching tree of vessels in the side of the basket. (B) Dextran was injected (15 uL over 28 s) at the anterior end of heart while heart pumped in the visceral direction. The image was obtained 6.5 min after end of injection. Dextran leaves the heart in two large vessels that then branch repeatedly to supply the visceral region, e.g., stomach and gonads. Some blood flows into the dorsal vessel and then moves toward the anterior. Scale bar at bottom left represents 10 mm.

In a second experiment, dextran was injected into a vessel at the anterior end of the heart pumping in the visceral phase. The dye was injected at the juncture of the two large vessels that leave the heart from the posterior end (Fig. 3B). One vessel bends sharply in the ventral direction and then branches to supply the stomach and gonads. The other vessel branches toward the intestine and the main dorsal vessel. The injection also inadvertently introduced fluorescent dextran into the pericardial sac that encompasses the inner heart tube. Thus the heart appears to be filled with fluorescent dextran even though at the time this image was obtained most of the fluorescent dextran in the interior heart tube has moved to the visceral region.

As with other ascidians, the heart consists of two concentric cylinders fused at the ends (Millar, 1953). The surface of the outer heart tube is relatively static, and thus does not directly participate in the pumping process. The pericardial space between the tubes is filled with fluid. Fluorescent dextran injected into the pericardial space of the C. inflata heart clearly defines the tube enclosing the heart (Fig. 4). Dextran remained in the pericardial space during a 24-hour period when the heart was beating at its normal rate. Thus, this space is quite isolated from the rest of the animal.

Figure 4 Fluorescent dextran injected into the pericardial space.

Image obtained more than 24 h after the injection and the heartbeat was normal throughout this period. Ventral is at bottom and anterior is to the right; the scale bar at bottom represents 10 mm.

Heart structure and function

The C. inflata heart reverses the direction of peristaltic contractions periodically as has been observed in all other tunicates. Constrictions traverse the heart in an average of 1.4 s and approximately every 4 min, or 180 beats, the direction of movement reverses (Table 1).

Table 1 Six consecutive heart beat phases of the peristaltic heart.

Visceral flow is from heart to viscera, branchial flow is from heart to the branchial basket.

Direction	Time (s)	Heart beats	Beats/s (Hz)	
Visceral	236	176	0.75	
Branchial	318	220	0.69	
Visceral	243	160	0.66	
Branchial	293	197	0.67	
Visceral	261	183	0.70	
Branchial	252	170	0.67	
Visceral: avg (std dev)	247 (13)	173 (12)	0.70 (0.05)	
Branchial: avg (std dev)	288 (33)	196 (25)	0.68 (0.01)	

The inner cylindrical tube contracts and twists progressively along its length to produce a moving constriction that forces the blood from one end to the other. The flow of stained blood cells enables movement of the inner heart tube to be followed in some detail. As one peristaltic constriction approaches the end of the heart, another constriction forms at the other end, and thus the heart tube is never open. The constriction that moves the blood through the heart tube does not appear cylindrically symmetric but is rather a twist in folds of the myocardium. About halfway along the heart tube is a stationary pinch of about half the average diameter. As the twist approaches this pinch, blood is forced through and a new fold appears on the other side with mirror image twist. Thus, the pinch does not appear to function as a valve. The anterior half of the heart, the heart-tube folds, and the vessel from the heart to the right side of the branchial basket is seen in Fig. 5A, and diagrammed in Fig. 5B. Video S2, which has the same field of view and orientation as Fig. 5, documents the movement of the heart myocardial folds.

Figure 5 Folds in the internal heart tube at the central constriction.

(A) is a photograph of the ventral end of the heart and the vessel that connects it to the right side of the branchial basket. The tunicate is rotated relative to previous figures, and the ventral edge is upward, anterior to the left. It has been stained with neutral red. (B) is a diagram of the same field with the viscera at upper right labeled. Segments of the heart tube containing blood on each side of the twist are labeled “H.” Scale bar represents 2 mm.

While rapidly moving stained blood cells can be seen in the heart, the high velocity generates streaks in video frames, and cells come in and out of focus due to the complex flow pattern. However, it was possible to follow the position of individual blood cells in a large vessel about 1 mm anterior to the heart as seen in Fig. 5, and thus to determine the velocity versus time profile (Fig. 6). Two cells were followed through two heart beats while they moved about 2 mm. The velocity versus time curves are approximately sinusoidal with a period slightly more than a second, consistent with Table 1. There is a reproducible small dip in the region of maximum velocity. The velocity is always positive, i.e., cells always move in the direction of the heartbeat, but the minimum velocity is only about a tenth the maximum value.

Figure 6 Velocities of blood cells in a large vessel.

A video with approximately the same field of view as Fig. 5 was disassembled into individual frames (30 per second), and the position of two densely stained cells was followed through sequential frames. Values in the graph are moving averages using Gaussian weighed values from five consecutive frames. The average blood cell velocities over the two complete heartbeats was approximately 0.9 mm/s.

In a typical animal (Fig. 1), the heart is about 12 mm long, and a constriction moves through the heart every 1.4 s, giving an average velocity of 9 mm s−1. The peak velocity of about 2 mm s−1 in the vessel just anterior to the heart (Fig. 6) is thus several-fold lower than the velocity of the heart constriction. However, this is but one of two large vessels the heart empties into, and the total cross sectional area of these vessels can only be crudely estimated. Blood cell velocities in three small vessels in the mantle at the extreme anterior end of the animal are much smaller than seen in the large vessel (Fig. 7). These velocities appear stochastic, with an average of about 0.3 mm s−1, but all are positive (move in the same direction) throughout several heartbeats.

Figure 7 Velocities of blood cells in three small vessels.

A video of stained blood cells in the mantle surrounding the exit siphon was analyzed in the same manner as described in Fig. 6.

Blood flow in the branchial basket

The cylindrical branchial basket occupies a major portion of the tunicate. As seen in Fig. 3A, blood flows from the ventral end of the heart to the middle of each side of the branchial basket. On the interior of the basket a rectangular network provides support and distributes blood (Fig. 8A). Blood then flows into an exterior layer perforated with pairs of spiral stigmata resembling the Ionic volutes of Greek columns (Fig. 8B). Cilia line the inside edges of the stigmata, but can only be seen by their motion. Fluorescent dextran injected into the blood reveals the structure of the duct containing the stigmata in low resolution (Fig. 8C). Blood cells could be seen moving in the area immediately adjacent to and between the stigmata. This is consistent with stigmata being openings through a double layer of cells, and blood circulating in the space between these cell layers.

Figure 8 Layers of the branchial basket.

The axis of the tunicate is horizontal with anterior to the right. Height of view in each panel is 1 mm. Each panel is from a different animal, but with corresponding positions and orientations. (A) Interior surface (stained with neutral red). The two prominent horizontal bands in the figure are the longitudinal bars of the basket. There are five vertical vessels perpendicular to each transverse bar that are under the longitudinal bars (only the two on the right are clearly defined in this image). (B) The exterior surface of the basket (stained with neutral red). The spirals that resemble Ionic volutes are the openings (stigmata) through which water is pumped. Higher magnification (not shown here) reveals the motion of beating cilia on the edge of the stigmata. (C) Exterior surface after injection of fluorescent dextran posterior to heart. Blood flows through a complex network of ducts linked together with net circulation perpendicular to the axis of the tunicate. Scale bar at bottom left represents 1 mm.

Blood from the branchial basket is collected by vessels along the ventral and dorsal edges of the branchial basket to flow to the posterior of the animal (summarized in the schematic of Fig. 2). When the heart reverses direction, the direction of blood flow in all parts of the animal reverses.

Discussion

Distribution of injected dextran and the flow of blood cells are complementary methods, dextran defining the larger vessels and their permeability while cell flow also reveals small vessels and blood velocity. Circulation of blood can be followed through most of the tunicate body.

Blood leaves the anterior end of the heart via two vessels supplying the two sides of the branchial basket. The blood then flows through transverse tubes into ducts containing the stigmata. Blood is collected by one large vessel on the ventral and another on dorsal side of the body to flow back into the visceral region. Blood is collected from the visceral region by a branched vascular tree merging into two large vessels entering the posterior end of the heart. It has not been possible to trace blood flow through the entire visceral region, specifically in the space between the posterior ends of the dorsal and ventral vessels and the ends of the vascular tree that feeds blood into the heart. The major difficulty is the density and opacity of the organs in this region. It is always possible that blood does not flow through clearly defined vessels in the viscera, but rather through a network of cavities as seen in the liver and spleen of vertebrates (Richardson & Granger, 1984).

Ruppert (1990) has described transport of sea water by the neural gland across the wall of the pharynx into the vasculature of the ascidian Ascidia interrupta, and suggested that the neural gland regulates vascular volume after contractions, “squirts.” However, Corella inflata does not contract often, and in the present investigations there were no contractions.

Vessel structure

The two large vessels that leave the anterior end of the heart follow asymmetric paths to the two sides of the branchial basket and do not appear to be associated with any of the symmetric structures in that region. In that sense they are independent structures.

The molecular integrity of vessels can be inferred by the rate infused fluorescent dextran leaks into surrounding tissue. A halo of fluorescence can be seen around the large vessels supplying blood to the branchial basket about 5 min after the start of the injection (Fig. 3A). However, since these vessels supply blood to smaller vessels much of the halo seen at this resolution could represent distribution, not leakage. Dextran flowing into the posterior region (Fig. 3B) remains in the major vessels for more than 6 min after the injection. Dextran of this molecular size has been used to follow plasma leakage in comparable times from vessels in the hamster cheek pouch induced by leukotrienes (Dahlen et al., 1981), and leakage could be compared in normal and treated animals. In the present report we can just show that tunicate vessels contain dextran for at least five minutes, a time comparable to vertebrate vessels. However, blood pressure in tunicates is approximately 100 times lower than that in vertebrates (Jones, 1985), thus absolute tunicate vessel permeability could well be much larger than found in vertebrates and still give comparable leakage rates.

Heart structure and function

The slightly curved cylindrical heart pericardium extends across most of the right side of the body at the anterior edge of the visceral region. While clearly visible on the right it is mostly obscured by the visceral mass when viewed from the left. Concentric with the pericardium is the inner myocardial heart tube, which contains the blood and joins with two large vessels at each end. If homologous with the heart of C. intestinalis, the myocardial tube is connected to the pericardial tube by a raphe which extends along the anterior length of both. A peristaltic heart branching into two vessels at each end, is the canonical structure of the vertebrate embryonic heart soon after the heart tube is formed (Gilbert, 2010; Santhanakrishnan & Miller, 2011).

The heart myocardium moves the blood using a constriction, or twist, that travels along the tube with a velocity of about 9 mm s−1. As one twist approaches one end of the tube a new twist forms at the other end, and thus there is always at least one twist to constrain blood flow. The shape of the myocardial heart tube between twists is an elongated ellipsoid which is distorted as it passes through the median constriction. Thus, when blood flows into vessels of fixed diameter at the ends of the heart its velocity is not constant, but rather has a quasi-sinusoidal shape similar to the profile of the blood bolus in the heart.

The performance of the tunicate peristaltic heart is quantitatively different from the “technical” peristaltic pumps used in industry, laboratories and hospitals (Manner, Wessel & Yelbuz, 2010). The flexible tubes of these pumps have a constant diameter which is compressed by a succession of rollers that are typically separated by several times the diameter of the tubing. Movement of the rollers then produces a constant flow rate that is only transiently modified when the roller leaves the tubing.

The pulsating movement of blood cells in the branchial phase of blood motion appears to be in phase throughout the animal, Video S1. This means that the pulse wave velocity (PWV) is large compared to the size of the animal and the heart beat rate (Zamir, 2000). The PWV has been studied in humans because of its potential for diagnosis of abnormal artery rigidity. PWV values in humans are in the range of 2–4 m/s (Yamashina et al., 2002), so that the pulse in even distant parts of the body are out of phase by only a fraction of a heart beat cycle. Since the tunicate is about 100 times smaller than a human, the diameters of the vessels are correspondingly smaller, and the heart rate is only 2–3 times as rapid, one would expect blood cell movement to be essentially synchronous throughout the system unless blood vessels were two orders of magnitude more flexible. It is thus not unexpected that blood cells pulse in synchrony throughout the tunicate, consistent with a closed, fairly rigid circulatory system.

In one of the two large vessel near the anterior end of the heart blood cells were observed to have a semi-sinusoidal velocity-time curve with a maximum of 2.2 mm s−1. The fish is perhaps the best vertebrate comparison, and a blood velocity of 1.7 mm s−1 has been measured in the aorta of a 5 day-post fertilization Zebrafish (Watkins et al., 2012). Zebrafish of this age are only about 2.5 mm long (Parichy et al., 2009) compared to a typical length of 30 mm for the Corella in this study, and blood velocity increases as the fish grows. However, blood velocities in the tunicate and fish are at least roughly comparable.

The peak velocities of blood in vessels proximal to the heart will not necessarily be equal to the velocity of the heart twist, because the total cross-sectional area of these vessels may not be equal to the maximal cross sectional area of the heart. However, it is difficult to measure the diameters of the heart and vessels with much accuracy since the edges do not appear sharp in the microscope. In addition, since the area of a tube is proportional to the square of its diameter, errors in area are twice those of the diameter measurements.

As vessels extend from the heart and repeatedly bifurcate the total cross section must increase, since a decrease in blood cell velocities is seen in videos. Since retention of fluorescent polymer indicates that the vasculature is not leaky, at least for periods of a few minutes, the decrease in cell velocities is just a consequence of conservation of blood volume. In a variety of biological circulatory systems an increase in total cross section after bifurcation of a vessel follows Murray’s law, which states that the sum of the cubes of diameters of the branches equals the cube of the diameter of the trunk vessel. In the very compact circulatory tree of the tunicate it seems likely that the velocity of blood decreases almost continuously from the time it leaves the heart. The velocities of individual blood cells graphed in Fig. 6A are not the velocities of blood flowing past a fixed position in a vessel, but rather a Lagrangian plot of the velocity of a cell as it travels along a flow path in a vasculature with a gradient of decreasing blood velocity. Thus the velocity of these cells is a function of both the phase of the heart beat and the position of the cell along the vessel. Each cell described by Fig. 6 is followed through two heart beats, and the peak velocity in the second beat always lower than that in the first, presumably because the cell is then farther away from the heart and the cross sectional area is larger. If the velocity of the cells is always decreasing, even the shapes of the velocity versus time curves graphed in Fig. 6 have been distorted compared to a curve measured at a fixed position. Thus averages of these peak velocities have little meaning since the cells are at different positions in the vessel. While videos of cell movement provide an exciting possibility to map blood flow throughout the animal, computer implemented image acquisition and analysis is clearly required for any comprehensive quantitative study.

In vessels distant from the heart with a small diameter, for example in the body wall, Fig. 6B, maximum velocities range from 0.7 to 0.5 mm s−1. Cell motion is often very irregular and velocities do not follow a smooth curve in synchrony with the heartbeat. This irregularity is not surprising since the cells are seen to frequently hit and sometimes momentarily stick to the vessel wall, also impeding movement of neighboring cells. This behavior is not unlike the flow of red blood cells through the capillaries of vertebrates and the velocity of blood cells is an unreliable measure of fluid velocity.

The myocardial tube has a fixed constriction approximately in the middle. A change in the twist of the heart tube as the peristaltic contraction passes through this constriction indicates it is a real anatomical feature. It does not function as a valve as blood cells pass through it freely throughout the heart cycle. Hecht (1918) reported seeing a constriction, or “node,” in the middle of the heart of Ascidia atra, a tunicate in the same order as C. inflata. He described the node in detail, and stated it was a “real landmark in the structure.” However, Goodbody, observing the heart of Corella willmeriana, which is likely to be the same species as C. inflata, also observed an apparent constriction in the heart but believed it was not a real anatomical structure but just a “shifting of the position the raphe,” (Goodbody, 1974). He referred to Hecht’s report and stated that Hecht also thought the node was just a shifting of the raphe. In my opinion this is not an accurate description of Hecht’s claims.

Microscope optics, illumination, and ascidian species used by Hecht, Goodbody, and the present author are all different, and the two previous authors did not stain the animals. However, I have seen the “node” in all my observations, and along with Hecht believe it is a “real landmark.” Since the velocity profile of blood cells in the vasculature is a reflection of the shape of the blood bolus in the heart I propose that the dip in velocity seen at the maximum in all the velocity profiles of Fig. 6 is a consequence of this constriction.

The constriction divides the heart in two sections, which might suggest either an evolution toward or regression from a two chambered heart homologous to the two chambered heart of the fish. Davidson et al. (2006) demonstrated a requirement for Ets1/2, a transcriptional effector of receptor tyrosine kinase signaling, in formation of the heart tube in C. intestinalis. They also constructed a transgenic C. intestinalis producing a constitutively activated form of the Ets1/2 gene product in heart and tail precursor cells. They found an “unexpected phenotype” was produced, an animal with a functional two compartment heart. Presumably it was unexpected because a mere increase in a factor generated a different structure, not just one of a different size or proportion. This suggests an unused, or underused genetic scaffold for a two chambered heart is endogenous to C. intestinalis, and the increase in Ets1/2 acted on this scaffold to produce the new heart structure. The same scaffold could well be present in C. inflata but in that environment produces a single chambered heart with a middle constriction.

Blood flow in the branchial basket

The peripheral wall of the branchial basket consists of two parallel layers of cells, penetrated by openings, the stigmata, through which water is pumped by a stack of seven layers of cilia bearing cells along the circumference of the stigmata (Burighel & Cloney, 1997). The wall of the branchial basket is thus homologous to the double-layered lamella (Olson, 2002), that constitutes the gills of fish. However, in the fish the layers are closer together, are held in place by pillar cells, and of course are not penetrated by stigmata.

Blood, supplied by transverse ducts, seen in Fig. 8A, flows between the walls of the basket, Fig. 8C, and is collected by vessels along the ventral and dorsal edges of the basket. Thus, the shapes of the stigmata, which are spiral in C. inflata, define not only the flow of water through the basket, but also the flow of blood within the walls of the branchial basket. In C. inflata the two vessels from the heart to the middle of the sides of the basket, the transverse ducts, and finally the hollow walls of the branchial basket and the stigmata should all be considered part of the pattern of blood circulation characteristic of the species.

Comparison with the circulatory system of Ciona intestinalis

In contrast to C. inflata, no constriction in the middle of the heart tube has been described in C. intestinalis, however it would be difficult to observe since the entire heart bends sharply in the middle more than 90°. The stigmata of C. intestinalis are thin rectangles aligned with the longitudinal bars, as contrasted to the double curved stigmata of C. inflata. Millar did not describe the double walls of the branchial basket in C. intestinalis, but may have just been unaware of this feature.

In Text-Fig. 6, pg 57 Millar (1953) shows blood flowing from the heart through the ventral endostyle, across the branchial basket, and back down the dorsal vessel to the visceral region, a sequence that suggests a circulation. He cites Skramlik (1929) as finding that there is “true circulation of the blood,” and does not state opposition to this conclusion.

In C. intestinalis blood flows from the anterior end of the heart directly into the vessel running along the ventral edge of the endostyle. In C. inflata it is possible that early in the metamorphosis from the tadpole the heart was also at the midline but had vessels established to the middle of each half of the branchial basket. The heart then migrated from midline to the right half of the animal, passing over the endostyle to its final position and dragging the left branchial vessel along with it. Berrill (1936) describes the extensive migration of the heart relative to the viscera during metamorphosis of several genera of Ascidians; unfortunately, Corella was not mentioned.

Open or closed, tidal or circulating

Discussion of blood circulation in tunicates is complicated by potential confusion between two distinct characteristics of a circulatory system and by the use of terms that are not neutral with respect to these characteristics.

In an open circulatory system blood is constrained to vessels for a relatively short distance after leaving the heart, while in the remaining part of the circulation it freely percolates through the cells. Since tissues are directly bathed in blood, it is typically called hemolymph. In a closed system blood is confined to a distinct vascular space. Thus the use of “blood” in the title of this report suggests the author believes circulation in C. inflata is closed and confined to vessels. However, while blood in a closed circulation is confined, water, blood proteins and white blood cells percolate at various rates through vessel walls to become lymph. Thus, open versus closed may best be considered a quantitative parameter describing the degree or rate blood is filtered before reaching the cells. In this report, the persistence of dextran in the vasculature of C. inflata is comparable to that seen in the closed circulation of vertebrates and thus circulation in this tunicate can also be described as closed.

The endothelial cells that line or constitute the blood vessels of vertebrates are considered the essential barrier between blood and lymph (Reiber & McGaw, 2009). Since most invertebrates do not have endothelial cells, and Millar reports that endothelial cells are found only near the heart in the tunicate C. intestinalis, one might conclude that circulation in tunicates must be open. However, Reiber & McGaw (2009) argue that the open-closed character of circulation is a continuum, note that some invertebrates that lack endothelial cells have a functional closed circulation, and that the invertebrate cephalopod molluscs have a vascular system lined with endothelial cells.

Circulation can be used as a generic term for any fluid flow. However, used more specifically in physiology blood circulation is a flow that moves around a circuit. Blood circulates through a network of circuits in vertebrates. The hearts of tunicates, including C. inflata, pump in one direction for a period and then reverse for a similar time. An early text on tunicates states: “blood does not go around a true circuit but shuttles back and forth through the heart” (Newman, 1939). This quote describes a tidal or reciprocating flow, where blood is pumped from one compartment to another and then back through the same path, not around a closed circuit.

However, observation of blood cells in C. inflata reveal they move in circuits, as seen in Video S1, although the flow in all the circuits change direction when the heart changes its pumping direction. The apparent circulation-tidal contradiction is resolved in the case of C. inflata by consideration of the volume of blood pumped by the heart during one directional phase. The inter heart tube has a diameter of about 1.6 mm and length of 12 mm, to give a volume of 24 mm3. While the heart reversal pattern is likely to be quantitatively dependent on age and size of the animal, water temperature and oxygen concentration, and other variables, the data presented in Table 1 is sufficient to rule out tidal flow. The average number of beats for one cycle is about 270, and thus the average volume pumped in one cycle would be 6,480 mm3 if the heart functioned as a piston moving down a tube with constant diameter. However, as described previously the space carrying the blood is more like an elongated oval between the two twists in the heart tube. Thus the actual volume pumped in one cycle is likely to be several-fold lower. The total volume over the 4 min the heart pumps in one direction is larger than the total volume of the animal (tissue and enclosed water), and thus much larger than the total volume of blood. Thus, blood must circulate in this tunicate in the strict sense, it just circulates in opposite directions during the two directional phases of heart action. A similar calculation and conclusion has been made by Kriebel (1968) for circulation in the tunicate Ciona intestinalis.

If reversal of blood flow is not required by the anatomy of the circulatory system, there may be a physiological advantage in periodic reversal. One possibility is that the gradient of oxygen and nutrients along the circulation pathway is sufficiently steep that tissues in the second half of the circulation loop do not receive optimal levels. Periodic reversal would provide a more uniform delivery, and Ruppert, Fox & Barns (2004), on page 946 suggest this possibility. In addition, the peristaltic heart could alter the ratio of beats in one versus the other direction, and thus compensate for changes in nutrient or oxygen supplies and demands. Despite some promising studies that suggest environmental modification of the reversal patterns, e.g., Ponec (1982), there is still no generally accepted model for the function of heart reversals.

Comparison to arthropods

The circulatory system of the invertebrate C. inflata tunicate can be compared to the invertebrate arthropods; aquatic crustaceans and terrestrial insects. In arthropods the heart typically consists of segments of the large dorsal vessel that sequentially contract (in peristalsis) to pump blood (hemolymph) from a sinus into vessels extending into each segment of the body. The blood then percolates through the body to return to the sinus (Brusca & Brusca, 2002). In contrast, in C. inflata the heart is a single tube which extends diagonally across the entire body. While there is a partial constriction in the middle of the heart, there are no vessels at that location that provide an exit or entrance for blood. When pumping in the branchial direction it obtains blood from a vascular tree that extends into the viscera, and delivers blood to both sides of the branchial basket via another vascular tree, then returning to the visceral region by dorsal and ventral vessels. There is no evidence of a sinus that supplies blood to the heart.

However, a similarity between C. inflata (all tunicates) and arthropods is a much lower density of cells in blood compared to vertebrates. The packed cell density in vertebrates is typically close to 30 percent, with most cells being erythrocytes that carry oxygen from air filled lungs to muscles preforming mechanical work. However, in both tunicates and arthropods the cell density in blood is about ten-fold lower. No oxygen-carrying molecule analogous to hemoglobin has been identified in the tunicates and thus the oxygen carried by blood is approximately the amount carried by water (Goodbody, 1974). However, the blood of many species of arthropods contains the oxygen binding protein hemocyanin.

In insects, air filled vessels, the tracheal system, supplies oxygen from the atmosphere directly to muscles. In the larger crustaceans hemolymph flows through body gills, often associated with appendages that move to increase both blood and water flow. In tunicates, the major cells performing mechanical work are the cilia surrounding the stigmata, which pump water through the branchial basket and the mucus feeding net. However, these cells are in direct contact with sea water, the source of oxygen for marine organisms.

Only the low blood cell density in C. inflata made it possible to determine position versus time for individual blood cells, as seen in Fig. 6. A high cell concentration, comparable to that seen in vertebrates, would present a confusing scene of overlapping cells frequently hiding each other as they changed depth in the ensemble.

Evolutionary relationships

DNA sequences of 18S rDNA genes suggest that the genera Ciona and Corella are as closely related as the two extremely similar Ciona species intestinalis and savignyi (Stach & Turbeville, 2002). However, the complete genome sequences for C. intestinalis and C. savignyi suggested that the these two tunicates have diverged from a common ancestor at a time approximately equal to the divergence of the chicken and human (Berna, Alvarez-Valin & D’Onofrio, 2009). Thus current analysis of DNA sequences is not particularly useful in understanding the differences in topology of the blood circulation between Ciona and Corella. We will need the complete sequence on C. inflata and better guidance on what specific parts of the genomes to compare to understand the genetic basis of the circulatory system.

A comprehensive histological study of the blood vessels in ascidians, e.g., C. inflata and C. intestinalis, would certainly extend our understanding of the circulatory systems of tunicates and thus the evolutionary relation to vertebrates. An obvious question is whether there is any histological or molecular difference between vessels anterior and posterior to the tunicate heart. If so, this would suggest that the common ancestor to both vertebrates and tunicates had a unidirectional blood flow. The fact that blood circulates in a closed system in tunicates as it does in vertebrates suggests but does not prove evolutionary similarity. The fact that the two circulatory systems are analogous does not necessarily mean they are homologous (Mayr, 2001).

Supplemental Information

Video S1 Blood flow in the right side of the tunicate as seen in Fig. 2

The heart tube is the vertical stripe about one-third from the left side of image but to the right of the dark viscera. Peristaltic contractions moving downward force blood cells into vessels leading to the two sides of the branchial basket. There is always a constriction (clear segment in the video) in the heart tube, and blood does not flow in the reverse direction during a heartbeat.

Click here for additional data file.

Video S2 Folds in the internal heart tube at the central constriction

The orientation and scale of this video is described in Fig. 5.

Click here for additional data file.

Additional Information and Declarations

Competing Interests

Author Contributions

Data Availability

The author declares there are no competing interest.

Michael W. Konrad conceived and designed the experiments, performed the experiments, analyzed the data, contributed reagents/materials/analysis tools, wrote the paper, prepared figures and/or tables, reviewed drafts of the paper.

The following information was supplied regarding data availability:

The research in this article did not generate, collect or analyse any raw data or code.

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
