# Peer review of "Blood circulation in the ascidian tunicate Corella inflata (Corellidae)"

_PeerJ, doi:10.7717/peerj.2771_

## Round 0.1 · original submission · Major Revisions

If you decide to revise the work, please submit a list of changes or a rebuttal against each point from attached reviewer's suggestion which is being raised when you submit the revised manuscript.

Reviewer 1 ·

Basic reporting

See comments for detailed

Experimental design

See comments for detailed

Validity of the findings

See comments for detailed

Additional comments

In this manuscript, the authors describe the circulatory system of an adult ascidian tunicate Corella inflata using fluorescent dye, neutral red and dextran. The results are novel, and I think it is suitable for Peerj. Yet the data and presentation have to be improved before the paper can be published. My major concerns are as follows:
The authors suggested that the circulatory system of C. inflata is closed based on the microscopy observation and the calculation of blood cell speed. However, the present evidence is not sufficient. The issue of open or closed circulatory system in ascidian population is a subject of much debate. The author need show much strong evidence on his opinion. For example, whether author can track an intact cycle of the specific blood cells quantitatively in the circulatory system? In another way, whether plasmid eletroporation in C. inflata is possible as it is in Ciona intestinalis? If the answer is “yes”, then it is feasible to introduce a GFP marker in blood cells probably using mesoderm driver to track the blood cells. These works will be helpful for the author to define the circulatory of C. inflata is open or closed.
Another interesting issue is the periodic heart reversal in C. inflata. It is an important characteristic of circulatory system in ascidian. However, the author didn’t emphasize this point in the part of “Results”. It is further worth to address the histological and cellular processes for heart reversal in this system. Whether the constriction in the middle of the heart plays a role in this process? What ‘s molecular basis for the constriction? Is it actomyosin mechanical-driving? The author can validate this simply using phalloidin and myosin antibody staining.

Other comments:
1. Fig 1. It will be helpful for the audiences to understand well if the author can provide a three dimensional schema to present the full view of C. inflata circulatory system. The separated right- and left-diagrams in the figures are not clear.
2. In figure2 why the author only show right side of the branchial basket?
3. Fig.5. I suggest the author show time-lapse snap of a movie to present the process of constriction.

Reviewer 2 ·

Basic reporting

The article meets PeerJ standards.

Experimental design

The article meets PeerJ standards.

Validity of the findings

The article meets PeerJ standards.

Additional comments

This manuscript provides an elegant description of the circulatory system of the tunicate Corella inflata. The process of blood circulation in this tunicate is elegantly described, the images are clear, and the author does an exemplary job detailing both the strengths and limitations of the approach. My comments on this manuscript are relatively minor, and are detailed below.

1. The primary weakness of the manuscript pertains to the quantitative aspects of the work. The heart rate is not clearly spelled out (for a single animal it is in table 1; consider using Hz), but there is no information on variance. How many animals were analyzed? What is the average contraction rate and variance? Furthermore, the measurements of flow were obtained using a very limited number of cells (for example, two in figure 6), and it is not clear how representative this flow velocity is for this species.
2. In the results it should be noted that the velocity of the cells likely underestimates the velocity of flow. This is alluded to in the discussion as pertains to the peripheral regions but it likely is true within the heart as well. Furthermore, it should be more clearly noted (if the author agrees) that the acceleration and deceleration of these cells is a reflection of the contraction and relaxation of the heart muscle.
3. The manuscript does a great job placing the data in the context of what we know about the heart of vertebrates, but I could not help think of important parallels with the heart of insects. Much like the data presented here, the heart of insects is also tubular and contracts in a peristaltic-like manner. Furthermore, the heart of many adult insects (for example, mosquitoes) reverses contraction direction (interestingly, the heart of many larvae do not). Perhaps a brief discussion of these parallels, in the context of the anatomy of the circulatory systems, would enhance this manuscript.
4. A video showing a heartbeat directional reversal would be useful.
5. The introduction and discussion are quite long-winded, and could be significantly shortened. For example, some paragraphs in the discussion are simply an extension of the results and do not place the data in the context of prior work (for example, lines 365-371, 396-401), and other paragraphs are somewhat superfluous (for example, lines 372-375). Because of the length of these sections, the reader can lose track of what is most important.
6. Line 177, I believe it is “lysosomes” and not “lysozymes”.

---

## Round 0.2 · Minor Revisions

The author is encouraged to revisit the comments of Reviewer 2 and to improve the manuscript from its original version according to that reviewers comments. We believe that additional attention to their prior comments would improve the paper significantly

Reviewer 1 ·

Basic reporting

Fine

Experimental design

Fine

Validity of the findings

Fine

Additional comments

The author explained the reasons that he could not perform some experiments at this moment. I understand and agree that the current version is suitable for PeerJ.

Reviewer 2 ·

Basic reporting

The manuscript conforms to PeerJ policies.

Experimental design

The experimental design is clearly described.

Validity of the findings

This manuscript is mostly unmodified from its original version.

Additional comments

This manuscript is mostly unmodified from its original version. The author missed the opportunity to improve the manuscript during the revision process.

---

## Round 0.3 · accepted · Accept

This manuscript is well modified from its previous version. And the current version is suitable for PeerJ.